

**Assessment of Forest Fire Rating Systems in Typical Mediterranean Forest, Crete, Greece.**
Mohamed Elhag*[1] and Silvena Boteva[2]
[1]Department of Hydrology and Water Resources Management, Faculty of Meteorology,
Environment & Arid Land Agriculture, King Abdulaziz University
Jeddah, 21589. Kingdom of Saudi Arabia.
[2]Department of Ecology and Environmental Protection, Faculty of Biology, Sofia University,
Sofia, 1164, Bulgaria.
*Correspondence to: melhag@kau.edu.sa
**Abstract**
The Fire Weather Index (FWI) module was tested under the Mediterranean- type conditions of
Crete (Greece) for the two fire seasons 2008-2009. High correlations were found between the
Fine Fuel Moisture Code (FFMC) and the Duff Moisture Code (DMC. The Drought Code (DC)
was insignificantly correlated with the soil moisture content. No significant correlation was
found between the area burned by wildfires and any component of the FWI system during the
studied period, unlike fire occurrence with which most of the components were highly
correlated. Meanwhile, the Keetch-Byram Drought Index (KBDI) of the American Forest Fire
Danger Rating System (NFFDRS) was also examined under the same conditions. It provided a
useful means of monitoring general wetting and drying cycles, but is inadequate for indicating
daily fire danger throughout the fire season in our region. Weak correlations between the KBDI-
the fire occurrence and the area burned were found for the two fire seasons studied-2008-2009.
Correlations between the KBDI and litter, duff and soil did not give statistically sound results.
On the contrary, the KBDI seemed to predict with high accuracy the moisture content of three
annual plants (*Piplatherum miliaceum, Parietaria diffusa, Avena sterillis*) with a shallow rooting
system of *Pinus halepensis* forest understory in the region. This indicated that the index was
adequate, to a certain extent, to represent the upper soil layers' water status, while it is unsuitable
to predict needles moisture content of *Pinus halepensis*, which has a deep rooting system.
**Keywords:** Danger Rating Systems, Data Analysis, Forecasting, Forest Fires, Fires Risk,
Moisture Content.



## 1. Introduction

Fire presents a main disturbance to natural ecosystems in the Mediterranean regions, leading to considerable ecological and economic losses. Thus, there is a great benefit can be obtained throughout anticipation. Therefore, the likelihood of fire incidence on a given date and its expected severity can be anticipated in advance. Fire danger rating is the expression of both variables and constants environmental factors that influence the occurrence, behavior, suppression efforts and detrimental effects of wildland fires in a certain area (Amatulli et al. 2013).

The incidence and the spread of forest fires be contingent on several environmental and anthropogenic factors. Even in regions like the Mediterranean region where most fires are iniciated by humans, natural circumstances that affect the fuel properties play a very significant role in the number of fires and the burned area (Bajocco et al. 2015).

Monitoring of live and dead fuel moisture can be used as an indicator of forest fire danger, if combined with simultaneous measurements of meteorological parameters. Forest fire danger rating systems are based on the integration of meteorological parameters with bio-physical characteristics of the fuels to forecast their combustibility and flammability, combined with the risk posed by human activities and natural phenomena (Pereira et al. 2011)

During the past six decades, much research has been accomplished and incorporated within several fire danger systems. These systems have evolved from several regional versions into the single National System of today. The Canadian Forest Fire Danger Rating System (CFFDRS), referred to prior to 1976 as the Canadian Forest Fire Behavior (or Behavior Rating) system, which embraces all aspects to evaluate fire danger and the forecast of fire behavior, including the



Fire Weather Index (FWI) System and Fire Behavior Prediction (FBP) System. (Pasqualini et al.

54  2011).

The CFFDRS has been under development in its present form since 1968, when the Canadian
Forestry Service (CFS) adopted a modular approach to a new National System of fire danger
rating. The first major module or sub-system of the CFFDRS, the (FWI) provides numerical
rating of relative wildland fire potential in a standard fuel type on level terrain (Dexter et al.
2013). The second major subsystem of the CFFDRS was conceived on the original modular
approach, as a series of regionally developed guides to fire behavior characteristics for specific
fuel complexes. These "Burning Indexes" or fire "behavior indexes" have become known as the
Canadian Forest Fire Behavior Prediction (FBP) System, Together with the (FWI) form the
National System of Fire Danger Rating System in Canada (Pasqualini et al. 2011).
While FWI was originally developed for use in Canadian pine forests (de Jong et al. 2016), it
was also extensively used in different environments and in other countries like France,
Indonesia, Malaysia, Mexico, New Zealand, Portugal, Spain, and the USA (Wang et al. 2017). It
was applied both for establishing new relationships between the FWI System components and
the fuel moisture/fire behaviour observed in local fuels and for distinguishing periods of high fire
activity. FWI index has also been found to outperform or at least match the performance of other
fire danger rating systems regarding highlighting periods with high fire activity in non-native
environments (de Jong et al. 2016). Furthermore, computer systems for fire management have
been used in Canada since the early 1970s. Later, in 1992, the Canadian Forest Services
investigated the application of geographic information systems for constructing the fire
management information systems and developed the spatial fire management system (Lawson &
Armitage 2008). In the 1980s and 1990s, these remote automatic weather stations and



respectively the associated communications technology allowed the collection of weather data
from isolated locations in almost real time on a local and even a national level (Taylor and
Alexander 2006). The CFFDRS is also used in the framework of Copernicus European Forest
Fire Information System (EFFIS) system which supports the services responsible for forests
protection against fires in the EU countries and providing current information on fires to the
European Commission services and the European Parliament.
In reviewing early fire danger research in Canada, two perceptions are worth to consider. First
the development process which was kept from system to system. Second, there was a simplicity
trend. Both are necessary for weather measurement and in the method of calculation (Wrathall

85    1985).

It is well understood that the incidence and behavior of forest fires depend mainly on short-term
weather influence of no more than several days duration (Amatulli et al. 2013). Yet, throughout
the history of fire danger rating, runs a persistent interest in the effects of weather over a much
longer term. Accounting for long-term drying is necessary because it provides guidance to the
fire manager during critical conditions. This does not imply that fires cannot occur without prior,
long-term moisture deficiency (Dimitrakopoulos & Bemmerzouk 2003); in other terms a drought
condition is not a prerequisite for the occurrence and spread of fire in any area (Amatulli et al.

93    2013).

The Keetch-Byram Drought Index (KBDI) was first introduced. It was defined as a number
representing the net effect of evapotranspiration and precipitation in producing cumulative
moisture deficiency in deep duff or upper soil layers. The material may be soil humus. It may
also be organic material that consists of buried wood, such as roots in varying degree of decay, at


different depths below the mineral soil surface. The KBDI is a daily index that ranges from 0 to
800, where higher values are associated with drought. The index is calculated from daily
observations of precipitation amount and maximum air temperature (Keetch & Byram 1968).
Conceptually, the index expresses, in hundredths of an inch, the moisture deficiency in soil, after
accounting for density of vegetation coyer, precipitation, and evapotranspiration losses. KBDI is
easy to compute and provides a continuous record because it is updated daily. It is good for
certain localities typical of fire sites as the index relies on point-source data. Since 1990, the
KBDI has been in use at selected Mediterranean locations of Greece (Dimitrakopoulos &
Bemmerzouk 2003). Also, Dolling et al. (2005) reported a strong link between the KBDI and
total acres burned for the Hawaiian Islands. KBDI is one of the several indicators that fire
managers use to track fire potential in Hawaii as it is used also by the Florida Division of
Forestry for classifying fire-season severity and is applied by fire managers in Texas to monitor
fire potential and effect burn bans (Dolling et al. 2009).
The objective of this study is to test and evaluate the following FFDRS, to propose possible
modifications that would better adapt these systems to the Mediterranean conditions. The
implemented forest fire danger rating systems are the Canadian Forest Fire Danger Rating
System (CFFDRS), and Keetch-Byram Drought Index (KBDI) of the American National Forest
Fire Danger Rating System (NFFDRS).
**2. Materials and Method**
**2.1. General description of the studied area**
Akrotiri region is situated 6 km East of Chania (Crete, Greece), at an altitude of 185 m a.s.l., at
approximately 35° 31' N and 24° 03' E. The mean annual rainfall is about 600 mm, December,



January and February being the period of the higher precipitations (Elhag & Bahrawi 2016). The
ecosystem investigated is a typical Mediterranean forest presented by *Pinus halepensis* with 50%
crown closure. The stand is southwest exposed with a slope of 20 %. The experiment was settled
for two consecutive fire seasons in 2008 and 2009. Daily measurements were conducted at
14.00h. Samples were taken from soil, litter, duff and under the canopy of the forest. They were
transported after sampling in hermetically sealed aluminum containers, for preventing
evaporation. For each parameter, three samples were collected. The daily moisture value is the
average of the three values. The daily weather data (air temperature, air relative humidity, 10m
wind speed, and 24h accumulated precipitation) for calculating the (FWI) indexes were obtained
from the Akrotiri Airport meteorological station, located about 5 km from the site where the
experimental site was established (Fig. 1).
**2.2. Canadian Forest Fire Danger Rating System**
The system is dependent only on weather and does not consider differences in risk, fuel, or
topography. It provides a uniform method of rating fire danger across Canada. The six
components are described below:
*2.2.1. Fine Fuel Moisture Code (FFMC)*
This code is an indicator of the relative ease of ignition and flammability of fine fuel. In order to
permit conversion of moisture content into a code, a scale, called the FF scale, was defined based
on the following assumptions and equations:
$$F = \frac{59.5(250 - m)}{147.2 + m}$$    Eq.1



Where F is the FFMC and m is the minimum moisture content (which was set equal to 2% in the
scale of the old FFMC: the "Tracer Index") of litter and other cured fuels.
### 2.2.2. Duff Moisture Code (DMC)
It was developed after (Van Wagner 1970), field work, mainly in *Pinus resinosa* and *Pinus*
*banksiana* stands. The method was based on transferring rectangles of organic matter to trays of
60 x 40 cm in an area set in forest floor, and to weigh daily. The type expression is:
$$P = c[\log(Mmax) - \log(M - E)]$$    Eq. 2
Where c is a scale constant, P is code value, M is moisture content, and E is the equilibrium
moisture content. According to the data recorded, M max was set equal to 300 and E to 20.
### 2.2.3. Drought Code (DC)
The Drought Code (DC) was improved by (Murray et al. 2012) and gives information on the soil
water rather than on the moisture condition of a given slow-drying forest fuel. Later on, several
researchers (Van Wagner 1970, Dimitrakopoulos & Bemmerzouk 2003) found that such an
index was quite suitable to represent certain fuels, since, like the soil layer, it loses moisture
exponentially. As an exponential expression of the moisture equivalent Q, the chosen scale
equation is:
$$D = 400 \ln(800/Q)$$    Eq. 3
Where, D is the current DC. The constant 400 represents the maximum theoretical moisture
content that can be held by the fuel represented by the DC.
### 2.2.4. Initial Spread Index (ISl)



The ISI is merely the product of functions of wind and fine fuel moisture, together with a
reference constant 0.0208. This constant was determined at a later stage, and was designed to
make the B-scale FWI equal to 40 for an arbitrary set of conditions. The constant itself was
multiplied by 10 to provide a convenient range of numerical values for the ISI. The equation is
then formulated as below:
$R = 0.208 f(W) f(F)$      Eq. 4
Where, R is the Initial Spread Index.
### 2.2.5. Buildup Index (BUI)
The BUI is a combination of the DMC and the DC. Since the DC was introduced to the system
after the DMC, a method was desired to give a limited, variable weight to the DC, reserving the
main effect to the DMC. When the DMC is near zero, the DC should not affect the daily fire
danger, no matter how high its level is. When the DMC and the DC are combined, the BUI is
given by:
$U = 0.8 PD/(P + 0.4D)$      Eq. 5
Where U is BUI, P is DMC, and D is DC.
### 2.2.6. Fire Weather Index (FWI)
The FWI is obtained by using the ISI and BUI values. To give the Fire Weather Index meaning
as a measure of fire intensity, factors are required for both rate of spread and fuel consumption.
The ISI clearly represents the rate of spread, but BUI is simply a blend of two fuel moisture
codes. The intermediate function required to calculate the FWI is expressed as follows:



$f(D) = 0.1Rf(D)$          Eq. 6
Where, R is the present day's ISI that is adjusted by the factor 0.1 to fit the scale. The following
relation can give the final equation:
$\ln S = 2.72(0.434 \ln B)^{0.647}$          Eq. 7
For general use, the FWI and its various system components are usually rounded to the nearest
whole number; any FWI value of less than 0.5 will, thus, be reported as zero.
**2.3. Keetch-Byram Drought Index**
The KBDI is a daily index that ranges from 0 to 800, where higher values are associated with
drought. The index is calculated from daily observations of precipitation amount and maximum
air temperature (Keetch & Byram 1968). Conceptually, the index expresses, in hundredths of an
inch, the moisture deficiency in soil, after accounting for density of vegetation coyer,
precipitation, and evapotranspiration losses.
If climate elements that affect transpiration, such as temperature, relative humidity, and solar
radiation have constant values and is observed a directly proportional rate between the amount of
water loss from the duff layer and its amount in the layer, then the conceptual equation can be:
$w = w_c exp(-\tau/t)$          Eq. 8
Where w is the available water (in inches) in the soil-duff layer for plants, wc is the
corresponding field capacity (in inches) of available water in the layer, $\tau$ is the time (in days)
when the soil-duff layer loses moisture, and t is the evapotranspiration timelag (in days)





necessary for the soil-duff layer moisture content values to decrease to 1/e of its initial value,
where e is the base of natural logarithms).
The next step is to establish a relationship among the evapotranspiration timelag t, the
temperature T and the mean annual rainfall R. The functional equation can be written as shown:
$1/t = f_1(T)f_2(R)$                                                                  Eq. 9
In which $f_1$ and $f_2$ are time functions that can remain undetermined. t is shown as a function of R
for two different values of T. Since there would be no vegetation for R = 0, it might appear that
t→∞ as R→0. Moreover, the moisture deficiency Q will be defined by:
$\log dQ = \log(w_c - Q) + \log f_1(T)\log f_2(R) + \log d\tau$                        Eq. 10
When determining the functions $f_1(T)$ and $f_2(R)$, it is required the evapotranspiration timelag t to
be written with more specific notation -$t_{TIR}$.
If one with T = T and the other with T = $T_0$ and with R having the same value in both equations,
then their ratio can be expressed as:
$\frac{t_{T,R}}{t_{T_0,R}}\bigg) = \left[\frac{dw_{T,R}}{dt}\bigg/\frac{dw_{T_0,R}}{dt}\right]^{-1}_{t=0}$     Eq. 11
It is needed to approximate the relationship obtained by equation (9) with an empirical equation
which is reasonable and consistent with the involved main physical concepts. Therefore, the
approximation of equation (9) leads to the following exponential equation:
$t_{T,R} - t_{T,\infty} = K\,exp(-aR)$                                                 Eq.12



In which a is a constant and K is a function of T only. However, when R = 0, then $t_{T,0} - t_{T,\infty}$
and therefore K = $t_{T,0} - t_{T,\infty}$.
Where, t as written with the appropriate subscripts for the specific notation form. When
comparing this equation with equation (9) it shows that $f_2(R)$ can be considered as the quantity:
$$f_2(R) = \left(t_{T_0,\infty} f_1(T_0)\right)[1 + y_0\, exp(-aR)])^{-1} \qquad \text{Eq. 13}$$
In the range from T = 50° F to T = 110° F, the empirical equation can closely approximate the
potential evapotranspiration rate curve:
$$-\left[\frac{dw_{T,50}}{dt}\right]_{t=0} = 0.352 \, exp\,(0.0486\,T) - 3.015 \qquad \text{Eq. 14}$$
Since the potential evapotranspiration ratio in the right part of this equation will be unchanged
for all values of R; it can be expressed with regard to equation (14). If we set the reference
temperature ($T_0$) at 80° F, equation (14) will have a numerical value for the potential
evapotranspiration rate amounting to 14.18 hundredths of an inch per day.
When T = $T_0$ = 80° F (arbitrarily chosen as the reference temperature) and wc = 800 hundredths
of an inch of water, then, from the above equation, $t_{80,50}$ = 56.41 days. Hence, from equation
(24), it follows that $t_{80,\infty}$ = 25.64 days. Expressing the terms of evaluated numerical constants
with only dQ in the left member gives the final equation:
$$dQ = \frac{[800 - Q][0.968 \, exp(0.0486T) - 8.3]\,dt}{1 + 10.88 exp(-0.0441R)} * 10^3 \qquad \text{Eq. 15}$$



The drought factor dQ, is conveniently computing daily in which case the time increment dt is
placed equal to 1 day. The final Spread Index unit equation can be written in the following
equation mentioned by Leverkus et al. (2014):
$$dQ = \frac{[203.2-Q][0.968\,exp(0.0875T+1.5552)-8.3]\,dt}{1+10.88exp(-0.001736R)} * 10^{-3}$$   Eq. 16
In the derivation of the basic equations, the fuel layer has been included with the soil. In the
setting of wc at 8 inches of water, it is assumed that the wc refers both to the soil and the fuel
layer.
**3. Results and Discussion**
**3.1. CFFDRS**
The burned area is the most obvious characteristics of a fire used in statistics. It can be dependent
on a number of factors mainly the occurrence of simultaneous fires, policies and priorities in
controlling fires, differences in fire accessibility, organization efficiency of the fire control,
composition and amount of fuel, weather conditions and topography.
**3.1.1. Burned Area, Number of Fires and the Components of the (FWI).**
The variables tested as predictors of area burned and number of fires are the three moisture codes
FFMC, DMC, and DC, the two intermediate indices, ISI and BUI, and the FWI. The correlation
matrix of these variables shows highly significant correlations between fire occurrence and DMC
(r = 0.89), DC (r = 0.78), BUI (r = 0.90), and FWI (r = 0.60). Despite these results, there is no
significant correlation between any component of the (FWI) system and burned area. A step-wise
multiple linear regression analysis was also performed but results indicated that the components





of the FWI explained an insignificant part of the variance in the burned area (Fernandes et al.

2014).

### 3.1.2. Fine Fuel Moisture Code (FFMC).

The potential range of the FFMC is from 0 to 101. In the given case, FFMC values range from
10.3 up to 96.5, with a mean value of 84.95 for the two investigated fire seasons. In this case
90% of the computed values are above 75, as 65 is the threshold value at which the likelihood of
fire ignition increases exponentially (Jimenez-Gonzalez et al. 2016). These observations suggest
that most of the days during the season are appropriate for fire occurence (Levin et al. 2016).

### 3.1.3. Duff Moisture Code (DMC).

The DMC has a virtually unlimited range. This implies that, given suitable fuel beds, the most of
the days during the fire season will have a potential for extreme fire behavior (Parr et al. 2007).

### 3.1.4. Drought Code (DC).

The DC responds slowly to environmental changes (Van Wagner 1970). In the spring, its initial
value depends upon the intensity of the previous fall and winter rains. The observation of the DC
values shows that the early spring value is close to 0, rising as the season progresses to values
which exceed 1360 in the early fall (the largest DC value of the two fire seasons was 1367,
recorded in October 2008). The average monthly rate of the DC shows an accelerating increase
in the early and late spring, and a slower increase during summer and a progressive decrease in
the early fall, after the first rains (Andrew et al. 2016).

### 3.1.5. Initial Spread Index (ISI).





Mean ISI values are generally small and relatively invariable in time. A combination of drying

weather and strong winds in late spring creates a maximum which then diminishes off to about

1/3 of its value by October. The highest mean and maximum values occur in June and July, with

the summer period also encountering also the maximum values with a mean of 14.65.

**3.1.6. The Fire Weather Index FWI.**

The FWI combines the values obtained from ISI and BUI (Ganteaume et al. 2013). The duration

of the fire season is variable from one year to the other are showed in Table 1. For the two fire

seasons, May represents the threshold month where FWI values are classified in the "High" and

"Extreme" classes in most days, thus indicating the actual start of the fire season. When

comparing the two studied fire seasons, the one in 2009 was much longer (May-November) than

that of 2008, when the fire season was restricted to May-September. This is due to the difference

in the amount of fall precipitation (120.6 mm for 2008 against 25.0 mm for 2009), which is

reflected in the FWI values. Consequently, it can be reported that the FWI can be successfully

used to indicate the beginning, the peak and the end of the fire season in the Mediterranean-type

ecosystem of Greece (Petropoulos et al. 2010, Turco et al. 2016).

The observation of the fire danger classes shows that the summer period (June- August) accounts

for about 95% of the "Extreme" values recorded during the whole fire season, thus indicating the

height of period with fires (Arpaci et al. 2013). May and September also experiences a big

percentage of FWI values in the "High" and "Extreme" classes. The analyses of the number of

fires in relation to FWI classes (Tab. 1) indicate clearly that more than 80% of fires occur during

days within the "Extreme" FWI class.



The statistics parameter indicates that measured (observed) and the predicted values of fine fuel
moisture are highly correlated. When conducting a t-test for both measured and predicted values
of litter moisture content it indicated that there was no significant difference at 95% confidence
level. Daily variations of moisture content of predicted and observed fine fuel is shown in Figure

2.

The two sets of points have similar trend during the investigated seasons. It is observed a better
correlation during the summer season, when it is not raining. This proves the model sensitivity to
daily changes in climate elements, namely temperature, relative humidity, and wind speed. In
occasions instantly after rain, this model under-predicts the moisture content slightly, but after
that resumes in a short time. Correlation between measured and predicted values (at 95%
confidence level) gave an r = 0.89.
**3.1.8. Duff Moisture Content.**
Even if a correlation at 95% level confidence between the measured and the predicted duff
values of moisture content gives an r = 0.75, the model predicts higher values for all descriptive
statistical parameters. Moreover, a t-test of paired predicted and observed values for duff
moisture content indicated significant difference at 95% confidence level.
The visual observation of comparison chart of predicted and observed duff moisture values (Fig.
3) shows that the two sets of points seems to behave differently. In the higher range of moisture
content, the predicted values present a delay in time before they start responding to rain
occurrence. This may be due to the torrential nature of precipitation in the Mediterranean region,
and/or to the discontinuous canopy closure characteristic of the Mediterranean pine forest type,
which differs from the Canadian conditions (Parr et al. 2007).



In addition, the duff layer is certainly notably less important in the Mediterranean conditions,
regarding quantity and depth, than that of the typical Canadian environment. In the lower scale,
the model seems to be limited in its predictions, as soon as the observed duff moisture content
values decrease fewer than 20%. This can be explained by the fact that the DMC was set with an
equilibrium moisture content of 20%. This means that the lowest scale predicted by the model
should be at least equal to 20. Unlike, during the summer season the duff moisture content, in our
ecosystem, is in most cases below the threshold of the model predictions.
**3.2. KBDI**
**3.2.1. Evaluation of the KBDl Trends with Time.**
The study shows a very high drought index, the mean value ± standard deviation was 660 ±
182.9 for the period March 2008 - November 2009 (640 days). In 66.1% of the days the drought
index exceeds the value 650 constituting the "Extreme drought" class. An annual trend can be
discernible, though the effects of year-to-year differences are non-negligible and the period
studied is certainly short. Following the end of the rainy period, usually at the beginning of
spring, there is a continuous increase in the drought index value, reaching quickly the maximum
value by mid of May. The precipitations occurring during the fall period with their small amount
and discontinuous feature seem to be insufficient to reduce significantly the drought index.
Consequently, the "Extreme" drought persists until the beginning of the winter time, where
relatively important and continuous precipitation occurrences combined with low temperature
drop the index value to the "High" and "Moderate" class. It is worth noting that the drought
index never restored the "Very Low" and "Low" classes. This is probably due to the
exceptionally drought weather that occurred during the year 2009.





The KBDI seems to over predict in time the end of the fire season. This emphasizes the fact that
long-term moisture deficiency cannot be used to forecast critical fire situations, because fires are
caused from a combination of factors that occur in conjunction with drought conditions.
Consequently, it is not possible to establish precise "threshold value" at which critical fire
situations may or may not develop (Dimitrakopoulos & Bemmerzouk 2003).
**3.2.2. The Keetch-Byram Drought Index as a predictor of foliage moisture content.**
Linear and exponential equations were tested for four species (*Avena sterilis*, *Parietaria diffusa*,
*Piplatherum miliaceum* and *Pinus halepensis*) to see which equation best fits the relationship of
plant moisture content and the KBDI (Tab. 2).
All species were linearly better correlated to the KBDl. *Piplatherum miliaceum* showed a strong
linear decay with increasing KBDI (meaning that the higher the KBDI and thus the more severe
the drought, the lower the plant moisture content); so, did *Avena sterrilis*, *Parietaria diffusa*, while
*Pinus halepensis* revealed a positive weak correlation.
Most the fluctuations in plant moisture content are accounted for by the KBDI for the species
monitored: *Piplatherum miliaceum* (Fig. 4) (variance explained 88%), *Parietaria diffusa* (Fig. 5)
(68%), and *Avena sterillis* (Fig. 6) (72%). For *Pinus halepensis*, the variance explained was very
low (4%). The T test was performed for the three-herbaceous species showing strong correlation
with the KBDI whose results indicated that the observed and the predicted (from the KBDI)
moisture content values are not significantly different at the 95% confidence level. On the other
hand, and according to Clavero et al. (2011), predictions of herbaceous plant and shrubs moisture
content within 20% are scientifically sound and adequate for prescribed fire planning (Brown et
al. 2015).



## 4. Conclusions


In the first part, the (FWI) system was tested against real data covering two fire seasons and can
be applicable as a method for meteorological fire risk assessment for the country. The FWI
supports to indicate the duration of the fire season, which is variable from one year to the other.
This highly risky period is generally confined between May and the end of September in Chania
region. The FWI indicates, also, with a relatively high accuracy the beginning, the peak, and the
end of the fire season. The analysis of the number of fires in its relation to FWI classes, for the
two fire seasons analyzed, revealed that about 95% of fires occur during "High" and "Extreme"
days. Highly significant correlations were found between "number of fires" and Duff Moisture
Content ($r = 0.89$), Drought Code ($r = 0.78$), Buildup Index ($r = 0.90$), and the Fire Weather
Index ($r = 0.60$). On the contrary, no significant correlation was found between the "burned area"
and any component of the (FWI). According to Alcañiz et al. (2016), this is not necessarily a
reflection on the accuracy and usefulness of the Fire Weather Indices. The analysis of longer
time series for further stations with similar environmental conditions to the one investigated
would bring more certainty about this specific point. In the second part, the KBDI was tested,
and showed to provide a useful means of monitoring general wetting and drying cycles, but was
inadequate for indicating daily fire danger throughout the fire season in Chania region. Weak
correlations between the KBDI and fire occurrence ($r = 0.24$) and the area burned ($r = 0.03$) were
found for the two fire seasons studied-2008-2009. This may be due to several reasons. The KBDI
supports to predict with high accuracy the moisture content of three annual plants (*Piplatherum*
*miliaceum, Parietaria diffusa, Avena sterillis*) with shallow rooting system representing the
understory of *Pinus halepensis* in Chania region. Separate models are developed for determining
their moisture content. This indicates that the index is adequate, to a certain extent, to represent



the upper soil layers' water status, while it shows to be inapt to predict the needles moisture
content of *Pinus halepensis*, which has a deep rooting system. The KBDI proved to be a
satisfactory way of monitoring general wetting and drying cycles, and thus warning fire
managers in the early stages about exceptionally wet and dry years. Furthermore, it is believed
that monitoring foliage moisture content of the main species in the Mediterranean region as
regard to their abundance and dominance and their involvement in most fires, and determining
the relationships with the KBDI. By applying the additional data received from KBDI, the EFFIS
system could use them for more accurate early fire warning and fire management planning such
as prescribed burning when conditions are convenient for it.

**Acknowledgment**


This article was funded by the Deanship of Scientific Research (DSR) at King Abdulaziz
University, Jeddah. The authors, therefore, acknowledged with thanks DSR for technical and
financial support.

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

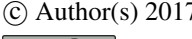



**List of Tables**
**Table 1. Percentage occurrence of FWI values within fire danger class intervals for**
**2008/2009.**

| FWI Limits Danger class | 0-1 Very Low | | 2-5 Low | | 6-12 Moderate | | 13-24 High | | > 24 Extreme | |
|---|---|---|---|---|---|---|---|---|---|---|
| | 2008 | 2009 | 2008 | 2009 | 2008 | 2009 | 2008 | 2009 | 2008 | 2009 |
| March | 25.8 | 29.03 | 51.6 | 29.03 | 16.1 | 25.8 | 6.45 | 16.1 | 0 | 0 |
| April | 3.3 | 16.66 | 13.3 | 3.33 | 50 | 23.33 | 30 | 33.33 | 3.3 | 23.33 |
| May | 0 | 0 | 0 | 3.22 | 38.7 | 0 | 39.03 | 6.45 | 32.25 | 90.32 |
| June | 0 | 0 | 0 | 0 | 0 | 6.66 | 10 | 0 | 90 | 93.33 |
| July | 0 | 0 | 0 | 0 | 0 | 0 | 3.2 | 0 | 96.8 | 100 |
| August | 0 | 0 | 0 | 0 | 0 | 0 | 3.2 | 0 | 96.8 | 100 |
| September | 0 | 0 | 0 | 0 | 0 | 10 | 13.330 | 0 | 86.66 | 90 |
| October | 19.35 | 3.22 | 16.12 | 0 | 32.25 | 9.67 | 19.35 | 22.58 | 12.9 | 64.51 |
| November | 23.33 | 13.33 | 16.66 | 10 | 40 | 13.33 | 16.66 | 20 | 3.3 | 43.33 |


**Table 2. Regression equations for estimating plant moisture content (x) from the KBDI (y).**

| Species | Equation | Regression Coefficient ($R^2$) | Variance explained ® |
|---|---|---|---|
| *Piplatherum miliaceum* | y = 904.23 - 1.881 * x | -0.93 | 0.88 |
| *Avena sterrilis* | y = 918.67 - 7.961 * x | -0.85 | 0.72 |
| *Parietaria diffusa* | y = 925.73 - 7.549 * x | -0.82 | 0.68 |
| *Pinus halepensis* | y = 511.78 - 1.934 * x | 0.20 | 0.04 |










**List of Figures**

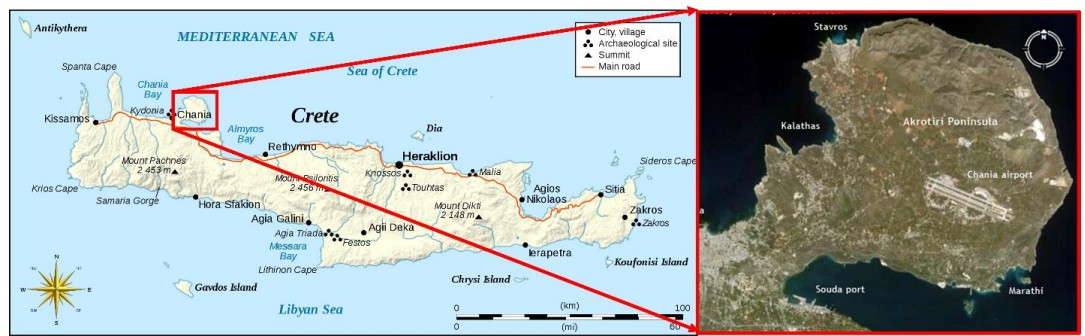


**Figure 1. Location of the study area.**

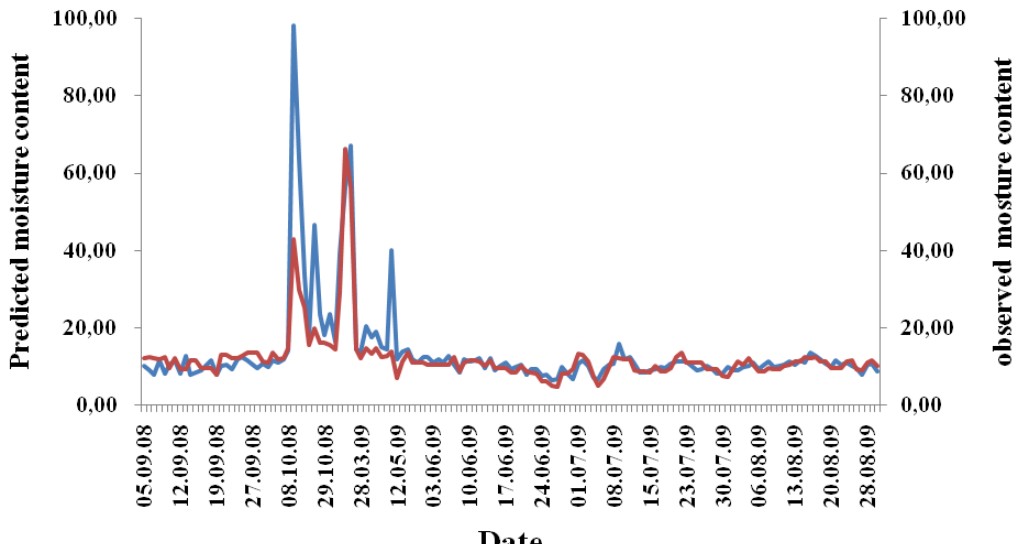


**Figure 2. Comparison of observed (red) and predicted (blue) fine fuel moisture content**
**values obtained from the (FWI) in Days.**




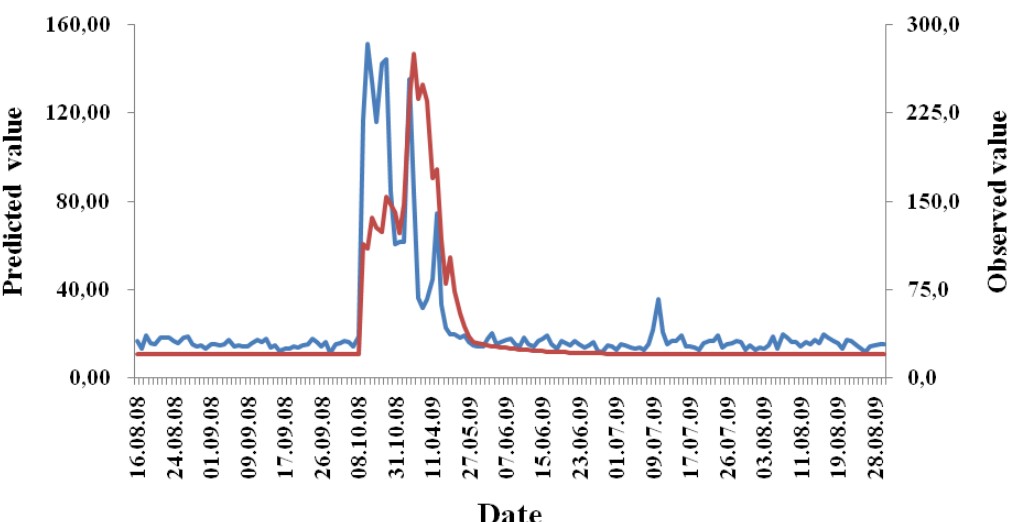


**Figure 3. Comparison of observed (red) and predicted (blue) duff moisture content values obtained from the (FWI) in Days.**

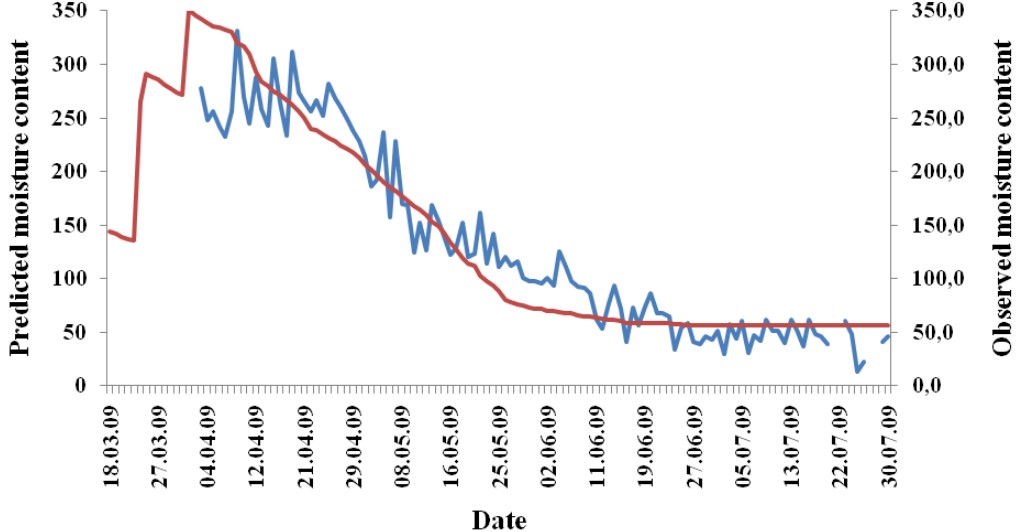


**Figure 4. Comparison charts of observed (dots) and predicted (line) moisture content of *Piplatherum milaceum* by the KBDI**



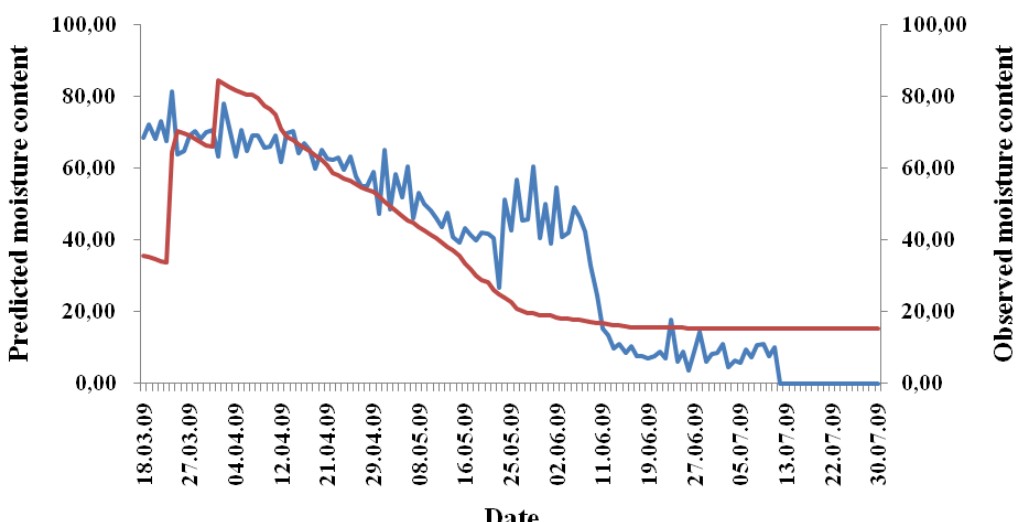


**Figure 5. Comparison charts of observed (Dots) and predicted (line) moisture content of**
***Parietaria diffusa* by the KBDI**


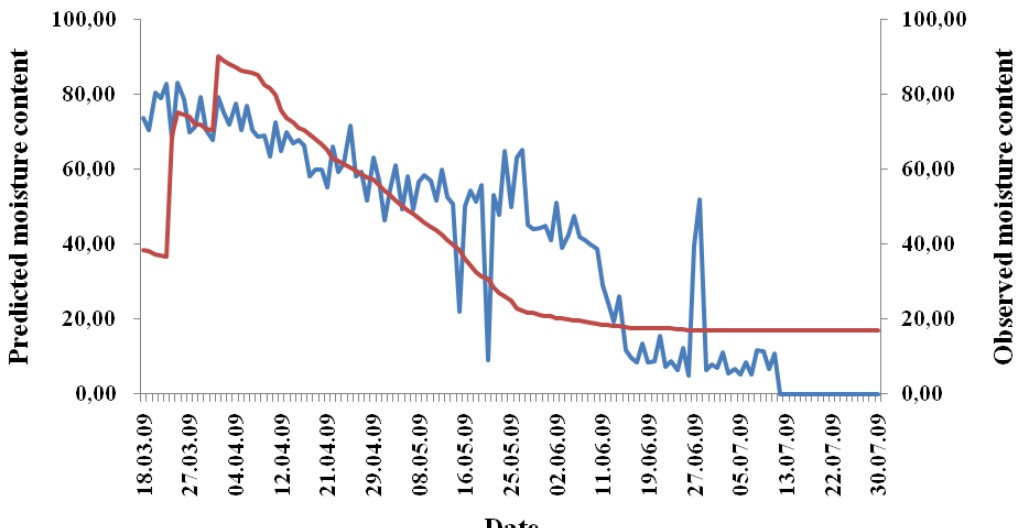


**Figure 6. Comparison charts of observed (Dots) and predicted (line) moisture content of**
***Avena sterillis* by the KBDI**
