# Peer review of "1. Introduction"

_Natural Hazards and Earth System Sciences, 2017_

## Short Comment (SC1) · 23 Oct 2017

Dear Editor, The authors were investigated the Fire Weather Index (FWI) module under the Mediterranean- type conditions of Crete (Greece) for the two fire seasons, in 2008 and 2009. In this study, the Canadian Forest Fire Danger Rating System (CFFDRS), and Keetch-Byram Drought Index (KBDI) of the American National Forest Fire Danger Rating System (NFFDR) were carried out. All indexes are explained in detail and the equations for each are given in the article. The advantages and disadvantages of the indexes were explained sufficiently. In the review report there are a few questions to the authors that I was wondering about the study. There are some

typographical errors in the text which needs correcting. I prepared a report for the author and uploaded to the system. Finally, in my opinion the article has a scientific content and it is suitable for publishing in your journal. Best regards, Dr. Nese YILMAZ

Please also note the supplement to this comment:
https://www.nat-hazards-earth-syst-sci-discuss.net/nhess-2017-318/nhess-2017-318-SC1-supplement.pdf

**Supplement:**

Dear Author,

I have some questions about your investigation. First, in my opinion, if you will give more information about the vegetation of the study region it will be better. It was used 4 annual plants for this work. Is there any reasons for selected this plants except their root types? Additionally, there are a few corrections below.

Yours Sincerely,

Dr. Nese YILMAZ

**COMMENTS:**

- ✓ There are a few typographical errors in the text which needs correcting. They are as follows:
1. In page 1, line 13 please correct the (DMC to (DMC).
2. In page 7, line 150 please correct the by (Murray et al. 2012) as Murray et al. 2012.

- ✓ In some equations, you need to explain this comments as follows:

1. In page 8, line 165, please explain what symbolizes the "W" and "F". in Eq.4.

---

## Referee Comment (RC1) · Anonymous Referee #1 · 28 Oct 2017

Dear Editors,

I am sending my review of the manuscript entitled Forest Fire Danger Rating Systems Assessment in the Mediterranean Type Environment, Crete, Greece. The manuscript concerns the problem with fires which is a very important problem due to their negative impact on nature and of course in humans leading to significant environmental and economic losses. The Introduction part describes in detail the concerned topic. The essence and use of CFFDRS and KBDI are described chronologically from their introduction until recent days. In Materials and Method part the authors give particular information about the applied indices supported with formulas and data that are

needed for the respective analyses. The part Results and discussion gives in detail the results obtained from the CFFDRS by using the respective codes and indices. The KBDI trends were evaluated with time and also it was tried to be used as a predictor for fires by identifying a relationship of KBDI and moisture content in plants typical for the investigated area - Mediterranean region. In the Conclusion part, the paragraph "This may be due to several reasons. The KBDI supports to predict with high accuracy the moisture content of three annual plants (Piplatherum miliaceum, Parietaria diffusa, Avena sterillis) with shallow rooting system representing the understory of Pinus halepensis in Chania region. Separate models are developed for determining their moisture content. "will be more suitable for the Discussion part. The manuscript is written in good English and is suitable for the Journal as it meets all the requirements. The research work has not only scientific but also applicable contribution given the problem concerned in it. Consequently, I sincerely encourage the authors to omit the redundant text in general and reduce the number of the equations included. The article is comprehensive and does it need a fill-up text. Based on the above said, I highly recommend this article to be published.

Yours sincerely,

---

## Referee Comment (RC2) · Anonymous Referee #2 · 30 Oct 2017

title: Assessment of Forest Fire Rating Systems in Typical Mediterranean Forest, Crete, Greece comments 1- the text need to be reduced 2-methodology should be shortened

---

## Author Comment (AC1) · 13 Dec 2017

Anonymous Referee #1 I sincerely encourage the authors to omit the redundant text in general and reduce the number of the equations included. Text is reduced but the questions are just satisfactory

---

## Author Comment (AC2) · 13 Dec 2017

The text needs to be reduced Seen and reduced Methodology should be shortened Methodology is just satisfactory

---

## Author Comment (AC3) · 13 Dec 2017

COMMENTS: There are a few typographical errors in the text which needs correcting. They are as follows: 1. In page 1, line 13 please corrects the (DMC to (DMC). Seen and corrected

2. In page 7, line 150 please correct the by (Murray et al. 2012) as Murray et al. 2012. Seen and edited In some equations, you need to explain these comments as follows: 1. In page 8, line 165, please explain what symbolizes the "W" and "F". in Eq.4. Seen and added

---

## Referee Comment (RC3) · Anonymous Referee #3 · 24 Feb 2018

Dear Editor, Here is my review for the manuscript entitled "Forest Fire Danger Rating Systems Assessment in the Mediterranean Type Environment, Crete, Greece"

As a general comment, the objective of the study (lines 111-112) " to test and evaluate the following FFDRS, to propose possible modifications that would better adapt these systems to the Mediterranean conditions" , cannot be supported by the analysis of the data, the results and the discussion. The authors provide some conclusions and recommendations, which cannot be justified in this manuscript , given the limited extent of the study and data, and also the lack of analysis based on important previous research and results related to the study of FWI in Mediterranean environments.

Below are some particular important comments: • The research paper of A. P. Dim-itrakopoulos, A. M. Bemmerzouk, I. D. Mitsopoulos (2010) entitled : "Evaluation of the Canadian fire weather index system in an eastern Mediterranean environment", which is a study the same area of Krete (Akrotiri), using very similar data and methodology, which showed the same results for FWI, has not been mentioned and taken into account. The above mentioned study, the conclusions and discussion should be considered by the authors as a basis for further evaluation of FWI.

• The classification system of FWI (the source is not mentioned by the authors) used in the current paper is not appropriate to the Mediterranean environment, as indicated by many previous studies and researchers. There exist relevant suggestions by the research community on the appropriate FWI classification including those EFFIS and Dimitrakopoulos et al. (in the research paper mentioned above).

• The following conclusions/suggestions for the use of the FWI and KBDI cannot be justified in the current study and should be avoided: - Lines 362-363: "In the first part, the (FWI) system was tested against real data covering two fire seasons and can be applicable as a method for meteorological fire risk assessment for the country." - Lines 390-392 :" The EFFIS system could use the additional data received from KBDI for more accurate early fire warning and fire management planning, such as prescribed burning when conditions are convenient for it."

Additional Comments • The authors do not provide information about the source, quality, amount of fire data used in their study. Thus, it is no clear how the results about the Burned Area, Number of Fires and the Components of the (FWI) and KBDI have been obtained (lines 247-250, 371, 377-378) • The meteorological station, which was the source of meteorological information for the evaluation of FWI and KBDI is at a rather long distance from the experimental site (5 km). The authors should justify the selection of the experimental site and/or the source of the meteo data.

Yours Sincerely,

---

## Author Comment (AC4) · 22 Apr 2018

Dear Editor, here is my review for the manuscript entitled "Forest Fire Danger Rating Systems Assessment in the Mediterranean Type Environment, Crete, Greece" As a general comment, the objective of the study (lines 111-112) "to test and evaluate the following FFDRS, to propose possible modifications that would better adapt these systems to the Mediterranean conditions" , cannot be supported by the analysis of the data, the results and the discussion.

This is a substantial assumption from the reviewer and unfortunately, it is not supported by any requested missing analysis

The authors provide some conclusions and recommendations, which cannot be justified in this manuscript, given the limited extent of the study and data, and also the lack of analysis based on important previous research and results related to the study of FWI in Mediterranean environments.

This is a substantial assumption from the reviewer and unfortunately, it is not supported by any missing justification. The paper presents a comparative evaluation method of two different indices, the FWI and the KBDI. The first one is widely used in all EU because it is a service of EFFIS (European Forest Fire Information Service), whereas the second one has rarely been used in the domain of forest fires in Europe because it is basically a drought index. Given the current debate about the formulation of new indices, the comparative evaluation of two indices, one of them not necessarily a fire danger one, seems rather a pioneer work and shall be considered.

Below are some particular important comments: âAËŸ c The research paper of A. P. Dim- ′ itrakopoulos, A. M. Bemmerzouk, I. D. Mitsopoulos (2010) entitled : "Evaluation of the Canadian fire weather index system in an eastern Mediterranean environment", which is a study the same area of Krete (Akrotiri), using very similar data and methodology, which showed the same results for FWI, has not been mentioned and taken into account.

We are fully aware of the mentioned article, to include it or exclude it from the citation is totally up to the authors, and it is not necessary to cite the article and even if the authors did not cite it does not mean that, the review allows him/herself to use a judgmental language. We decided not to include it as we are not in a position to judge the others scholarly work. We have our concerns regarding the mentioned article specifically the use of KBDI as a component of fire danger assessment, but not as a stand-alone fire danger index. Furthermore, in the comparative evaluation the Authors should put in evidence that KBDI 's relationship to fire danger is only that as the index value increases, the vegetation is subjected to increased stress due to moisture deficiency.

The above mentioned study, the conclusions and discussion should be considered by the authors as a basis for further evaluation of FWI. âAËŸ c The classification system of FWI (the source is not mentioned by the authors) used ' in the current paper is not appropriate to the Mediterranean environment, as indicated by many previous studies and researchers. There exist relevant suggestions by the research community on the appropriate FWI classification including those EFFIS and Dimitrakopoulos et al. (in the research paper mentioned above).

Generic personal point of view not supported with why is not appropriate. It's not a must that all scholarly work must not agree on everything. This is still scientific research. We have our own concerns not to use the above-mentioned article because of the observation about the beginning, the peak, and the end of the fire season is well coherent with EFFIS values. Unfortunately, this comparison was not been done because EFFIS decadal experience has not been considered, as already observed.

âAËŸ c The following conclusions/suggestions for the use of the FWI and KBDI cannot be ' justified in the current study and should be avoided: - Lines 362-363: "In the first part, the (FWI) system was tested against real data covering two fire seasons and can be applicable as a method for meteorological fire risk assessment for the country." - Lines 390-392 :" The EFFIS system could use the additional data received from KBDI for more accurate early fire warning and fire management planning, such as prescribed burning when conditions are convenient for it."

Generic personal point of view not supported with any requested missing justification. Due to its nature of cumulative index, the value of the KBDI increases and varies very slowly during the year, in short, it has a very low, and delayed sensitivity to weather variables, which reduces the interest of its use as a predictor of fire danger, which is a rather dynamic reality.

Additional Comments âAËŸ c The authors do not provide information about the source, ' quality, amount of fire data used in their study. Thus, it is no clear how the results

about the Burned Area, Number of Fires and the Components of the (FWI) and KBDI have been obtained (lines 247-250, 371, 377-378) The authors preferred to retain text consistency rather than adding metadata âAËŸ c The meteorological station, ′ which was the source of meteorological information for the evaluation of FWI and KBDI is at a rather long distance from the experimental site (5 km). The authors should justify the selection of the experimental site and/or the source of the meteo data.

This is the closet metrological station with complete data availability. In Creta there are 7 official metrological stations (Souda, Rethimno, Tymbaky, Iraklio, Kastelli, Ierapetra, Sitia), in addition to 10 Personal Weather Stations (PWS are individually-owned outdoor instruments that measure weather conditions, which they are unreliable) but all of them functioning perfectly and 6 of those stations are apart from the study area.

Yours Sincerely

---

## Referee Comment (RC4) · Anonymous Referee #1 · 24 Apr 2018

Dear Editor Dear Authors Dear Reviewers

I can see great efforts paid by all parties. The article is solid and worth fast publication. Moreover, the comments are useful to improve the context of the article. On the other hand, I must agree that the reviewer #3 comments are not totally clear and his/her language shall be revised in indeed. I encourage the editor to proceed with the article based on its valuable scientific content.

Yours

2017-318, 2017.

---

## Referee Comment (RC5) · Anonymous Referee #4 · 17 Jun 2018

Dear editors, authors and reviewers, The paper exmaines the applicability of the Canadian FFDRS and US KBDI as indicators of fire potential in Mediterranean forests, a useful goal given the fire prone nature of many Mediterranean vegetation communities. Some work is required on the paper, however, prior to publication in my view. The authors state (in lines 286-288 and 363) that the FWI is applicable as an assessment of fire risk in Crete. The study period covers only two fire seasons, however, one of which the authors note (lines 337-338) was a year of exceptional drought. I think that a longer period of study would be required to definitively assert the value of the FWI. An appropriate conclusion, I suggest, would be that the FWI shows promise and that a longer analysis is justified. To further support the author's case, it would be useful, as

reviewer 3 notes, for readers to be able to see the extent of fire activity during the two fire seasons, and understand how typical that activity was through the authors providing details of average fire numbers and area burnt. It isn't clear to me that the authors acknowledge that the KBDI was developed as simply an indicator of long-term dryness, and not specifically a fire danger index. KBDI doesn't, for example, include any dependence on wind or relative humidity which, depending on the type of fuels, affect fire behaviour to a greater or lesser extent, and which the FWI does incorporate. The only discussion that relates to this point is in lines 192-194, implicitly, where constant values of weather parameters are assumed - an assumption which, incidentally is not justified in the text - and in line 377, where the authors report that KBDI is not adequate for indicating daily fire danger. Related to this point is the assertion (lines 87-88) that forest fire activity is dependent mainly on short-term weather. It is true that short-term weather is important , of course, but in many forests antecedent conditions are also important for sufficient drying to have occurred to permit fuels to burn. The reason that indices such as KBDI were developed was to quantify this long-term drying. The assertion needs to be substantially qualified, or removed. I note reviewer #3's reference to an earlier paper, covering similar but not identical material, and agree that it would be very worthwhile, indeed important, to cite that paper. Finally, substantial additional editing is required, I suggest, prior to publication. I offer some examples but these are not by any means comprehensive: Lines 187-191 are a word-for word repeat of lines 98-102, including he mis-spelling of "cover"; line 235 refers to "initial Spread Index unit equation". The discussion here is about the KBDI, not the ISI (unless I've completely misunderstood the derivation, in which case greater clarity of argument is indicated); In section 3.1.4, it is not clear to me what is a result and what is being reported from the existing literature; Line 384 should read "... it is shown to be inappropriate to predict the needle moisture content..."; Lines 387-390 do not constitute a sentence. In summary, I concur with reviewer #3 that substantial reworking of the manuscript is required prior to publication. Kind regards,

---

## Author Comment (AC5) · 19 Jun 2018

Re: Reviwer#4 Dear editors, authors and reviewers, the paper examines the applicability of the Canadian FFDRS and US KBDI as indicators of fire potential in Mediterranean forests, a useful goal given the fire prone nature of many Mediterranean vegetation communities. Some work is required on the paper, however, prior to publication in my view. The authors state (in lines 286-288 and 363) that the FWI is applicable as an assessment of fire risk in Crete. The study period covers only two fire seasons, however, one of which the authors note (lines 337-338) was a year of exceptional drought. I think that a longer period of study would be required to definitively assert the value of

the FWI. An appropriate conclusion, I suggest, would be that the FWI shows promise and that a longer analysis is justified. To further support the author's case, it would be useful, as reviewer 3 notes, for readers to be able to see the extent of fire activity during the two fire seasons and understand how typical that activity was through the authors providing details of average fire numbers and area burnt. Acknowledged and edited to be "FWI shows promise" in the conclusion section. Regarding methodology, two fire seasons are enough to reach such conclusion. This is not the first work to endorse such results with two fire seasons only. Other scholarly work like Hély et al. 2001 and Dimitrakopoulos et al. 2011 reported the same in two seasons.

It isn't clear to me that the authors acknowledge that the KBDI was developed as simply an indicator of long-term dryness, and not specifically a fire danger index. KBDI doesn't, for example, include any dependence on wind or relative humidity which, depending on the type of fuels, affect fire behavior to a greater or lesser extent, and which the FWI does incorporate. KBDI is widely considered as a dryness index rather than fire danger index in many works of literature such as Srinivasan et al. 1998; Svoboda et al. 2002; Heim and Richard 2002 and Garcia-Prats et al. 2015. Therefore, authors considered KBDI as a dryness index.

The only discussion that relates to this point is in lines 192-194, implicitly, where constant values of weather parameters are assumed - an assumption which, incidentally is not justified in the text - and in line 377, where the authors report that KBDI is not adequate for indicating daily fire danger. Related to this point is the assertion (lines 87-88) that forest fire activity is dependent mainly on short-term weather. It is true that short-term weather is important, of course, but in many forests antecedent conditions are also important for sufficient drying to have occurred to permit fuels to burn. The reason that indices such as KBDI were developed was to quantify this long-term drying. The assertion needs to be substantially qualified or removed. It was not totally unjustified, Dimitrakopoulos et al. 2011 support the short-term weather. The reference is now incorporated in the text and the assertion is removed.

I note reviewer #3's reference to an earlier paper, covering similar but not identical material, and agree that it would be very worthwhile, indeed important, to cite that paper. Acknowledged and cited.

Finally, substantial additional editing is required, I suggest, prior to publication. I offer some examples, but these are not by any means comprehensive: Lines 187-191 are a word-for word repeat of lines 98-102, including the mis-spelling of "cover"; line 235 refers to "initial Spread Index unit equation". Acknowledged and redundant text is omitted.

The discussion here is about the KBDI, not the ISI (unless I've completely misunderstood the derivation, in which case greater clarity of argument is indicated); In section 3.1.4, it is not clear to me what is a result and what is being reported from the existing literature; Line 384 should read "... it is shown to be inappropriate to predict the needle moisture content..."; Lines 387-390 do not constitute a sentence. The discussion is about the KBDI, but it doesn't mean that ISI is ignored. Estimation of the DC is an ancillary procedure but worth to mention to give the readers the full picture of implemented methodology. Lines 384 is edited and lines 387-390 are removed.

In summary, I concur with reviewer #3 that substantial reworking of the manuscript is required prior to publication. Kind regards,

We thank reviewer #4 of his/her comments and his/her insights to enrich the current work but we, unfortunately, won't be able to do any extra work regarding the fire seasons and data collection. As you may have noticed, this is a completely independent work with no finical support from any kind or even a postgraduate program. Fieldwork and data collection was the worst part of this article especially when you take into consideration the data inconsistencies and discrepancies in Greek authorities. If review #3 and reviewer #4 insists on extra work, then authors kindly ask the reviewers to secure a source of fund to go on with extra work. We appreciate reviewer concerns, but we can also fulfill their concerns with our own personal money. Two fire seasons

are enough to reach the current conclusions and as we mentioned before this is not the first work to be based on two fire seasons only.

Please also note the supplement to this comment:
https://www.nat-hazards-earth-syst-sci-discuss.net/nhess-2017-318/nhess-2017-318-AC5-supplement.pdf

―――――――――――――――――――

---

## Author Comment (AC6) · 19 Jun 2018

your comments are highly appreciated
* * *

---

## Author Comment (AC7) · 22 Jun 2018

the comments are well received and followed literally according to the attached files. kindly appreciate the changes in the revised version